

# Larger lake outbursts despite glacier thinning at ice-dammed Desolation Lake, Alaska

Natalie Lützow[1], Bretwood Higman[2], Martin Truffer[3], Bodo Bookhagen[4], Friedrich Knuth[5], Oliver Korup[1,4], Katie E. Hughes[6], Marten Geertsema[7], John J. Clague[8], and Georg Veh[1]

[1]Institute of Environmental Science and Geography, University of Potsdam, Potsdam-Golm, 14476, Germany
[2]Ground Truth Alaska, Seldovia, AK, USA
[3]Geophysical Institute and Department of Physics, University of Alaska Fairbanks, Fairbanks, AK, USA
[4]Institute of Geosciences, University of Potsdam, Potsdam-Golm, 14476, Germany
[5]University of Washington, Civil and Environmental Engineering, Seattle, WA, USA
[6]Victoria University of Wellington, Wellington, New Zeal and
[7]Ministry of Forests, Prince George, BC, Canada
[8]Department of Earth Sciences, Simon Fraser University, Burnaby, BC, Canada

*Correspondence to*: Natalie Lützow (natalie.luetzow@uni-potsdam.de)

**Abstract.**

Many glaciers dam lakes at their margins that can drain suddenly. Due to downwasting of these glacier dams, the magnitude of glacier lake outburst floods may change. Judging from repeat satellite observations, most ice-dammed lakes with repeated outbursts have decreased in area, volume, and flood size. Yet, we find that some lakes oppose this trend by releasing progressively larger volumes over time, and elevating downstream hazards. One of these exceptions is Desolation Lake, southeastern Alaska, having drained at least 48 times since 1972 with progressively larger volumes despite the surface lowering of the local ice dam. Here we focus on explaining its unusual record of lake outbursts using estimates of flood volumes, lake levels, and glacier elevation based on a time series of elevation models and satellite images spanning five decades. We find that the lake grew by ~10 km$^2$ during our study period, more than any other ice-dammed lake with reported outbursts in Alaska. The associated flood volumes tripled from 200 - 300 $\times$ 10$^6$ m$^3$ in the 1980s up to ~700 $\times$ 10$^6$ m$^3$ in the 2010s, which is more than five times the regional median of reported flood volumes from ice-dammed lakes. Yet, Lituya Glacier, which dams the lake, had a median surface lowering of ~50 m between 1977 and 2019 and the annual maximum lake levels dropped by 110 m since 1985, to a level of 202 m a.s.l. in 2022. We explain the contrasting trend of growing lake volume and glacier surface lowering in terms of the topographic and glacial setting of Desolation Lake. The lake lies in a narrow valley in contact with another valley glacier, Fairweather Glacier, at its far end. During our study period, the ice front of the Fairweather Glacier receded rapidly, creating new space that allowed the lake to expand laterally and accumulate a growing volume of water. We argue that the growth of ice-dammed lakes with outburst activity is controlled more by 1) the potential for lateral expansion and 2) meltwater input due to ablation at the glacier front, than by overall mass loss across the entire glacier surface. Lateral lake expansion and frontal glacier ablation can lead to larger lake outbursts even if ablation of the overall glacier surface





accelerates and the maximum lake level drops. Identifying valleys with hazardous ice-topographic conditions can help prevent some of the catastrophic damage that ice dam failures have caused in past decades.

## 1 Introduction

Glacier thinning has accelerated in most mountain regions on Earth over the past two decades and has been particularly pronounced in the coastal regions of northwestern North America (i.e. Canada and Alaska). Between 2000 and 2019, glaciers in Alaska lost on average 66.7 Gt yr$^{-1}$ of ice, accounting for a quarter of the total global glacier mass loss outside the three ice sheets (Hugonnet et al., 2021).

Glaciers are an important buffer in the hydrological cycle as they intercept and store water for many decades (Jansson et al., 2003). Changes in seasonal meltwater regimes modulate river discharge and affect fluvial transport rates, sedimentation, and freshwater availability, which are crucial for land use and water resource management downstream (Willis and Bonvin, 1995; Moore et al., 2008; Huss and Hock, 2018). In coastal areas, glacier runoff affects water temperature, salinity, turbidity, and nutrient supply, impacting local marine ecosystems and wildlife habitats (Mernild et al., 2015; Arimitsu et al., 2016). Glaciers can also temporarily trap melt- and rainwater at their margins, forming lakes behind moraines, in bedrock depressions carved by glacial erosion, or behind the glacier body itself (Benn and Evans, 2010; Otto, 2019). Glacier lakes have attracted growing interest in research on natural hazard and risk appraisals because their dams can be unstable and fail suddenly with catastrophic consequences (Carrivick and Tweed, 2016; Zheng et al., 2021). Furthermore, glacier lakes are subject to mass movements that can result in landslide-generated tsunamis (Vilca et al., 2021; Geertsema et al., 2022; Lemaire et al., 2024). In populated areas, glacier lake outburst floods (GLOFs) have led to fatalities and socio-economic losses by destroying houses, infrastructure, as well as farmland, forests, and livestock (Carrivick and Tweed, 2016; Hock et al., 2019). The number and size of glacier lakes globally increased by about 54% and 11%, respectively, between 1990 and 2020 in response to atmospheric warming and glacier retreat (Zhang et al., 2024). In this period, Alaska had the highest regional increase in total glacier-lake volume; however, the number and area of ice-dammed lakes, which are sources of the largest reported glacier flood in this region, decreased (Rick et al., 2022, 2023; Veh et al., 2023).

Mechanisms leading to drainage of an ice-dammed reservoir include the enlargement of glacial conduits, overflow of the glacier, or flotation of the ice dam (Walder and Costa, 1996; Tweed and Russell, 1999; Huss et al., 2007; Kienholz et al., 2020; Clague and O'Connor, 2021). En- or subglacial tunnels may enlarge by ice melt due to frictional heat of the water flow or mechanical erosion (Liestøl, 1956; Mathews, 1965; Nye, 1976; Clague and O'Connor, 2021). In case of both flotation or overflow of the ice dam, the escaping water may enter a subglacial conduit once an existing or new inlet is reached or created (Anderson et al., 2003; Kienholz et al., 2020). Flotation of the ice dam is assumed to occur when the water depth at the dam reaches ~90% of the dam thickness (Thorarinsson, 1939, 1953), although this ratio may vary depending on the density of the ice dam. For instance, the ratio could increase if the ice dam contains much debris, and decrease if the dam is highly crevassed (Thorarinsson, 1939; Tweed, 2000). The outflow of an ice-dammed lake may end when the tunnel closes due to ice deformation





or mechanical collapse, or once the water supply is exhausted due to complete drainage of the reservoir (e.g., Clarke, 1982; Sturm and Benson, 1985). Once partially or fully drained, an ice-dammed lake can fill again and enter a cycle of periodic drainage, thus posing a repeated hazard downstream (Marcus, 1960; Post and Mayo, 1971; Mathews and Clague, 1993; Evans and Clague, 1994; Geertsema and Clague, 2005; Carrivick and Tweed, 2016; Otto, 2019). This 'jökulhlaup cycle' may cease when water cannot be impounded by the ice barrier due to downwasting and weakening of the ice dam or formation of a
permanent drainage pathway (Evans and Clague, 1994).

Concurrent with global trends, the timing and magnitude of ice-dam failures in Alaska has changed in recent decades (Rick et al., 2023; Veh et al., 2023). Judging from repeat satellite observations, ice-dam failures now occur earlier in the year and most lakes are smaller in both area and volume than in the past (Rick et al., 2023; Veh et al., 2023). Despite the global increase in glacier lake volume, this trend towards smaller lakes and floods is also observed for most single ice-dammed lakes
with recurring outburst floods and has been attributed to thinning of the local ice dam, limiting the storing capacity of the lake (Evans and Clague, 1994; Tweed and Russell, 1999; Geertsema and Clague, 2005; Shugar et al., 2020; Zhang et al., 2024). Examples include Hidden Creek Lake, dammed by Kennicott Glacier in Alaska, which lost 45% of its area between 2000 and 2019 while the glacier dam lowered by 1.6 m on average. In the same period, Tulsequah Lake, B.C., Canada shrank by 34% concurrent with a dam lowering of 4.9 m (Veh et al., 2023). However, a compilation of GLOFs throughout Alaska shows that
some ice-dammed lakes grew in size and hence potentially have drained larger volumes (**Fig. S1**). One of these exceptions is Desolation Lake in coastal southeast Alaska, which shows the largest growth regionally despite frequent, yet previously unreported, drainages in the past five decades. To date, processes leading to progressively larger GLOFs from ice-dammed lakes are not well understood and thus warrant more research. Our aim is to investigate the response of Desolation Lake to local changes in glacier size and mass that might explain its atypical pattern. We compile and analyse a 50-year time series of
digital elevation models, satellite images, and field data, including the first glacier survey conducted on Lituya Glacier, which dams the lake. We show how the lake formed and evolved and track changes in its size and drainage characteristics using time-stamped data from the ice dam.

## 2 Study Area

Alaska currently hosts 667 glacier lakes >0.05 km$^2$ (2016-2019), 62 of which are ice-dammed (Rick et al., 2022).
Since the 19$^{th}$ century, about 1800 outbursts from 127 ice-dammed lakes have been documented in Alaska (Lützow and Veh, 2024). Desolation Lake (58.765°N, 137.627°W) in southern coastal Alaska is dammed by Lituya Glacier, which flows from the Fairweather Range into Desolation Valley where it separates into two ice tongues, one damming Desolation Lake on the northwest and the other flowing southeast towards Lituya Bay (**Fig. 1a,b**; Ward and Day, 2010). Lituya Glacier has an area of 77 km$^2$ (RGI Consortium, 2017), of which about 13% is covered by rocks and debris (Scherler et al., 2018). Areas with exposed
ice are heavily crevassed (**Fig. 6a,d**). The glacier area in Desolation Valley that dams the lake is in the ablation zone with only





seasonal snow cover. Desolation Lake is further bordered by Fairweather Glacier to the northwest and Desolation Glacier flowing into the valley from the northeast (**Fig. 1b**).

The lake axis follows that of Desolation Valley and the NW-SE-striking, tectonically active Fairweather Fault. In 1958, before modern Desolation Lake began to form (**Fig.1d-f**), a magnitude 7.9 earthquake triggered a landslide that entered

Lituya Bay in front of Lituya Glacier, which was a tidewater glacier at that time (Miller, 1960; Lander, 1996; Mader and Gittings, 2002). The landslide triggered a tsunami that reached up to 524 m above sea level on the opposite hillslope and travelled the length of the bay and into the open Pacific Ocean. The run-up remains the highest of any reported tsunami worldwide (Mader and Gittings, 2002; Miller, 1960).

Desolation Lake is surrounded by steep valley walls with an average slope of ~32° (ArcticDEM 2020-09-11; **Table**

**S1**). Field observations, satellite imagery, and elevation models show that frequent mass movements of different size occur on these slopes. Since the onset of the GLOF cycle, sediment carried to the terminus of Lituya Glacier has formed a rapidly advancing proglacial delta, which today separates the glacier from Lituya Bay. Accordingly, Lituya Glacier ceased to be a tidewater glacier and became a landlocked valley glacier between 1990 and 1992. Crillon Glacier flows into Lituya Bay south of the head of the bay. A lake dammed by Crillon Glacier ~10 km from its terminus (**Fig. 1a**) has had no reported outbursts.

**3 Data and Methods**

**3.1 Outburst chronology and lake area mapping**

We reconstructed the outburst chronology of Desolation Lake and Lituya Glacier between 1882 and 1969 from historic air photos taken by Austin Post and from historic maps (see **Supplementary Table S1**). From satellite images acquired between 1972 and 2023, we determined the occurrence and timing of outbursts from Desolation Lake based on evidence of

sudden lake-level lowering and increases in proglacial sediment outwash carried into Lituya Bay. We obtained images from the Landsat 1 and 2 missions at 80-m pixel resolution and Landsat 3 at 40-m pixel resolution between 1972 and 1983 through the USGS Earth Explorer (https://earthexplorer.usgs.gov/, last access: 01 December 2023). For all subsequent years, we used Landsat 5, 7, and 8 data at 30-m pixel resolution obtained through the Google Earth Engine Platform (https://developers.google.com/earth-engine/datasets/catalog/landsat, last access: 01 December 2023). For the period since

2008, the image time series was further augmented by Planetscope, RapidEye, and Sentinel-2 data accessed through the Planet Explorer (https://www.planet.com/products/explorer/, last access: 01 December 2023). These products offer higher pixel resolutions of 3, 5, and 10 m, respectively. To quantify lake-area changes, we manually mapped the lake outlines using the last available cloud-free image before and the first image after each outburst identified from the satellite images with QGIS (*version 3.30.0;* retrieved from https://qgis.org/download/). In cases where the lake was partially covered by floating ice, we

mapped the lake shore at the position of the calving fronts.





## 3.2 Outburst volume and lake level estimation

We estimated outburst volumes in the programming environment R (*version 4.2.2;* https://cran.r-project.org/src/base/R-4/), using the mapped lake area outlines and the 2-m resolution ArcticDEM from 2020-09-11. This DEM (digital elevation model) shows the lake at the minimum water level at 197 m h.a.e. (height above WGS84 ellipsoid) and has the smallest glacier extent within the available ArcticDEM time series (**Table S1**). In our study area, the WGS84 ellipsoidal height is approximately +7 m compared to the mean sea level (EGM96 geoid). We flattened the lake surface area to a constant elevation of 197 m h.a.e. to remove floating ice captured in the 2020-09-11 ArcticDEM (**Fig. S2**). To estimate lake levels before and after each GLOF, we sampled the elevation of the DEMs along the mapped polygon outlines at 10-m horizontal spacing. The samples were only collected from the shorelines of the southern part of the lake, which is less steep than the northern part and therefore better captures lateral lake-level changes following outbursts (**Fig. S2**). We then filled the elevation model of Desolation Valley to the modal elevation of these point samples, creating a DEM with a flat surface that provides estimates of the water level before and after the GLOF. We used the $25^{th}$ and $75^{th}$ percentiles of the sampled elevation as an uncertainty measure of lake elevation (**Fig. S2**). For each outburst, we calculated the per-pixel elevation difference between the DEMs before and after the outburst. We then approximate the flood volume released from the subaerial parts of the lake as the sum of all pixel differences multiplied by the pixel area ($4$ m$^2$). We only estimated flood volumes for outbursts since 1985, when the meltwater trapped in crevasses has formed a coherent subaerial water body in contact with the valley walls. Based on the variance of the sampled lake elevations, we conservatively estimate uncertainties in the flood volumes of ±20%. Between 2016 and 2023, there are no elevation measurements for post-flood levels below the 2020 ArcticDEM lake level, hindering the approximation of outburst volumes.

## 3.3 Glacier elevation

We determined elevation changes over Lituya Glacier using four ArcticDEMs derived from Maxar WorldView 1, 2, and 3 satellite imagery acquired on 2013-11-08, 2016-12-10, 2018-09-02, and 2019-04-01 (**Table S1**). The ArcticDEM products have reported horizontal and vertical accuracy of ~4 m (Porter et al., 2022). To extend this time series, we generated a DEM derived from 21 historical aerial photographs acquired on 1977-09-01 as part of the North American Glacier Aerial Photography (NAGAP) project (**Table S1**). The 1977 NAGAP DEM was created following the image standardization and stereo-reconstruction methods described in Knuth et al. (2023) and using the AgiSoft Metashape photogrammetric software (*version 1.6.0*; http://www.agisoft.com/downloads/installer/). All DEMs were co-registered to the 2013 ArcticDEM and projected in the EPSG:3413 CRS prior to computing the per-pixel difference. To co-register the four ArcticDEMs, we used the Nuth and Kääb co-registration method implemented in the *demcoreg* Python library (Nuth and Kääb, 2011; Shean et al., 2016). Prior to co-registering the ArcticDEMs, we excluded unstable surfaces in the 2013 ArcticDEM by manually masking out the proglacial delta, glacier, water-covered surfaces, slopes steeper than 30° (**Fig. S3**), as well as a region of instability located northeast of the proglacial delta (**Fig. 4, S3**). To remove a residual tilt and co-register the 1977 NAGAP DEM to the



2013 Arctic reference DEM, we masked the glacier, proglacial delta, and water-covered surfaces, both in the 2013 Arctic reference DEM and the 1977 NAGAP DEM. We then transformed both DEMs to X, Y, Z point clouds and performed a co-registration using the iterative closest point (ICP) point-to-plane approach (Chen and Medioni, 1992) implemented in the *open3D* Python software (Zhou et al., 2018). Finally, we applied the derived transformation matrix to the unmasked 1977 point cloud. We calculated the vertical 1977-2013 cloud-to-cloud distances based on the 2013 ArcticDEM cloud using Cloud Compare (*version 2.12.3*; https://www.danielgm.net/cc/).

We assessed the accuracy of DEM and point cloud co-registration with histograms of the per-pixel or point-elevation changes between the DEM dates ($t_2$-$t_1$) within the unmasked areas, which assume to be stable surfaces with 0 m elevation change over time (**Fig. S4**). We further estimated the total elevation-change error ($\sigma_{\Delta h}$), as described by Shean et al. (2020), as the root-mean-square error over these surfaces from a random $\sigma_{\Delta h_{random}}$ and a systematic $\sigma_{\Delta h_{systematic}}$ error component:

$$\sigma_{\Delta h} = \sqrt{\sigma_{\Delta h_{random}}{}^2 + \sigma_{\Delta h_{systematic}}{}^2} \qquad (1)$$

We used the normalized median absolute deviation (NMAD) (Höhle and Höhle, 2009) of elevation change for the $\sigma_{\Delta h_{random}}$ component, representing the spread of noise across the stable surfaces and the mean elevation change, representing the systematic (local) elevation change bias, as the $\sigma_{\Delta h_{systematic}}$ component:

$$\sigma_{\Delta h} = \sqrt{NMAD(\Delta h)^2 + mean(\Delta h)^2} \qquad (2)$$

Lituya Glacier elevation changes were calculated within the reference glacier outline on 2013-11-08 with an area of 10.6 km$^2$, covering the part of the ablation zone of the glacier that dams Desolation Lake, in the following referred to as Lituya Glacier dam. This outline was cropped above the ice divide to ensure equal coverage of the elevation data (**Fig 4**). For all mass loss ($\Delta M$) estimates, we assume that all glacier elevation changes are attributed to the loss of glacier ice with a density ($\rho$ of 900 kg m$^{-3}$ (Huss, 2013). We estimated the loss of ice volume within the 2013 reference polygon by multiplying the polygon area () with the median elevation change ($\Delta h_{0.5}$). We estimated the total mass loss error ($\sigma_{\Delta M}$) from two normalized error components; the total elevation change error ($\sigma_{\Delta h}$) derived from equation (2), and the mapping error of the 2013 glacier outline ), (adjusted from Shean et al., 2020):

$$\sigma_{\Delta M} = |\Delta M| \times \sqrt{\left(\frac{\sigma_{\Delta h}}{\Delta h_{0.5}}\right)^2 + \left(\frac{\sigma_{\Delta A}}{A}\right)^2} \qquad (3)$$

Following previous studies we assumed a $\sigma_{\Delta A}$ of 10% (Kääb et al., 2012; Shean et al., 2020).

### 3.4 Ice-penetrating radar survey of Lituya Glacier

In June 2023, we measured ice thickness and determined bedrock elevation along two transverse profiles on lower Lituya Glacier (**Fig. 5b,c**), using an impulse radar system (Kentech) with ~2.5 MHz antennas and a custom built receiver. The rough debris-covered surface and frequent crevasses made a continuous profile impossible. Instead we conducted a series of point surveys, spaced at approximately 30 m, by leap-frogging transmitter and receiver antennas. The antennas were oriented



perpendicular to the survey profile. These point measurements were assembled into two profiles and return times picked manually. In some instances, more than one bed return could be identified. We used the two-way travel time and an assumed

wave speed of 169 m/µs to plot return ellipses, assuming an in-plane reflector. We interpret the envelope of these ellipses as the glacier bed. Elevations for the glacier bed were obtained by differencing the measured ice thickness from the 2019 ArcticDEM elevations (**Table S1**) of the glacier surface. Uncertainties in ice thickness stem from possible deviation from the assumed radar wave speed ($\pm 2$ m/µs), from the accuracy with which a return can be picked (about 0.1 µs), and from the interpolation from the discrete point measurements. Based on cross-over analysis from similar surveys, we estimate

uncertainties to be $\pm 20$ m (e.g., Tober et al., 2023).

**Fig 1.: Location and formation of Desolation Lake. a**; Oblique aerial view into Desolation Valley from the head of Lituya Bay. Desolation Lake is only partially filled. White arrows indicate the location of the two flood outlets. **b**; Location map and glacial setting of Desolation





Lake. **c-j**; Time series of historic maps (**c,d**; U.S. Geological Survey, 1951, 1961) and Landsat images (**e-j**; derived from the USGS Earth Explorer and Google Earth Engine Data Catalog) showing the formation and growth of Desolation Lake between 1951 and 2020. The yellow outlines in **g-j** show the mapped lake areas ($a_L$). The photograph by Austin Post inset into **d** is a view into Desolation Valley and Lituya Glacier from the head of Lituya Bay in 1969, shortly before modern Desolation Lake formed.

## 4 Results

### 4.1 Lake formation and outburst history

Desolation Lake began to form in the late 1960s when meltwater accumulated in longitudinal crevasses of Desolation and Lituya glaciers. This meltwater is visible in topographic maps and air photos of Austin Post in 1963 and 1969 (**Fig. 1d**). In the early 1980s, a single coherent water body formed and started growing on the surface of Desolation and Lituya glaciers (**Fig. 1e,f**). In the following years, the lake expanded towards rapidly retreating Desolation and Fairweather glaciers in the northwest (**Fig. 1g,h**). In 2013, Desolation Glacier had retreated so far that it no longer blocked the valley. This allowed Desolation Lake to double its surface area by connecting to meltwater trapped in crevasses in the northwestern part of the valley (**Fig. 1i,j; Fig. 2a**). In 2014, Desolation and Fairweather glaciers separated, leaving only floating ice on the lake. Until then, Desolation Lake had grown by about 10 km$^2$ within 35 years to a total area of 12.7 km$^2$ (**Fig. 1, Fig. 2a**). Since 2014, the maximum annual area of the lake decreased by about 4 km$^2$ to 8.6 km$^2$ in 2023 (**Fig. 2a**).

At least 48 outbursts interrupted the growth of Desolation Lake (**Table S2**), though none of these emptied the lake completely. Most outbursts occurred between June and September, but a few also happened between autumn and spring. Estimating the timing of GLOFs in winter is challenging because clouds and ice commonly cover Desolation Lake for several months, while the solar illumination, and thus contrast in shadowed regions, decreases. Since 1985, Desolation Lake has burst every year except for 1988, 1991, 1998, 2001, 2008, and 2011, when no drainage events were detected. Water was released two or more times in ten years of our survey period, usually first between July and August, followed by a second outburst in autumn or during the winter (**Table S2**). In 1999 and 2005, Desolation Lake partially drained at least three times.

### 4.2 Flood volumes

The minimum estimated outburst volumes of Desolation Lake range from $33 \pm 7 \times 10^6$ m$^3$ to $625 \pm 125 \times 10^6$ m$^3$ (**Fig. 2b**). Over the 30 years for which we have reliable estimates (1985-2014), flood volumes have at least tripled, with volumes between 200 and $300 \times 10^6$ m$^3$ in the late 1980s and up to $\sim 700 \times 10^6$ m$^3$ in the early 2010s (**Fig. 2c**). Flood volumes increased when the lake doubled its area by expanding towards Fairweather Glacier between 2006 and 2014 (**Fig. 1h-j; Fig. 2b**). This disintegration of Fairweather Glacier had already started in the 1980s when water-filled crevasses began to form and increase in number and size. Even before the lake claimed the area north of Desolation Glacier, the water from these crevasses drained during outbursts. We were unable to estimate the additional volume of water discharged from these crevasses, so our total estimated flood volumes are minima for this period. The lowest flood volumes were associated with floods in 2005 and 2006, just before this large-scale expansion. This two-year period had at least five detected partial drainages.





### 4.3 Lake-level changes

While flood volumes increased, pre- and post-flood lake levels of Desolation Lake fell substantially (**Fig. 2c**). The lake level before the 2022 GLOF was 207 m h.a.e., about 110 to 130 m lower than the pre-GLOF lake levels in the late 1980s. The post-flood levels decreased in a similar manner and dropped below the DEM-derived lake level of 197 m h.a.e. in 2016 (**Fig. 2c**). Satellite images show that the lake continued to decrease in width in the following years, indicating that the post-flood lake levels have further dropped since then (**Fig. S5**). In 2023, the last year of our record, both pre- and post-flood lake levels remained below the DEM lake level of 197 m h.a.e..





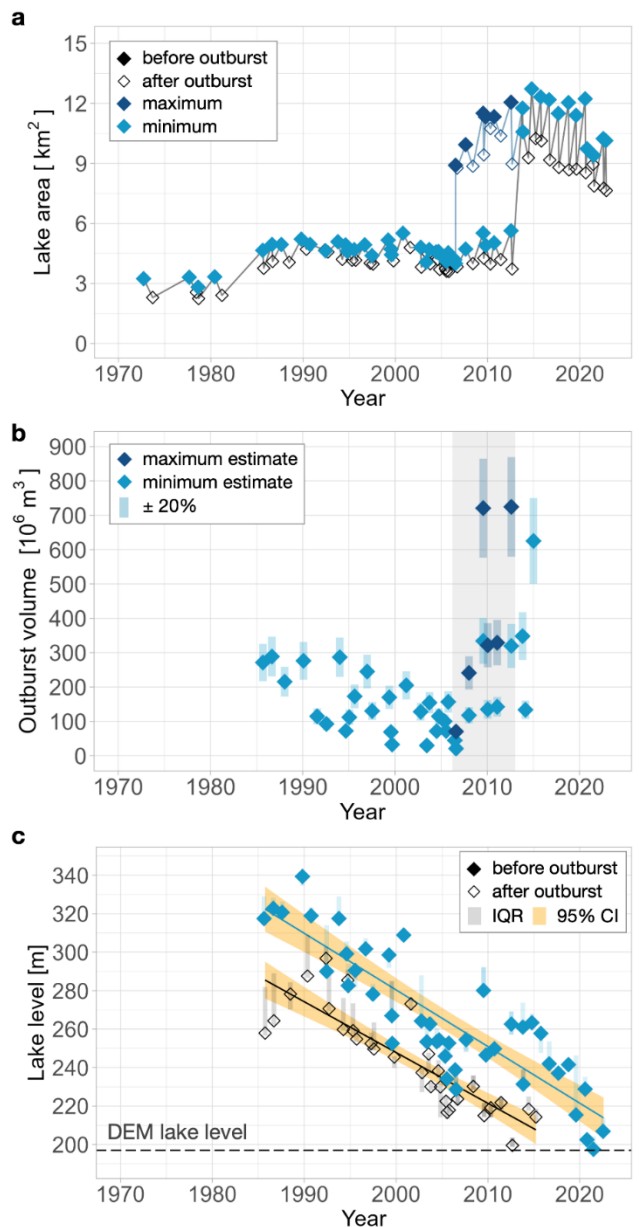

**Fig. 2: Temporal changes in lake size, outburst volumes, and lake levels from 1972 to 2023. a**; Lake area before and after the flood. **b;**
Outburst volumes with error bars assuming an uncertainty of ±20%. In **a** and **b**, data for GLOFs between 2006 and 2012 are colour-coded
to minimum and maximum estimates from mapped lake extents, constrained by the uncertain lake shores along rapidly melting Fairweather
Glacier (this interval is grey shaded in **b**). **c;** Lake level before and after GLOFs. The blue and black lines are linear regression models fitted
to pre- and post-outburst levels, including the 95% confidence interval (CI; orange). The horizontal dashed line is the lower limit of the
volume calculation (see section **3.2**). Pale blue and grey vertical intervals for outbursts mark the 25th and 75th percentiles (IQR) of the lake
elevations extracted from the DEM. Diamonds in **a** and **c** bracket the timing of the outburst, with filled diamonds (open) representing the
last available image before (first available image after) each GLOF.



Diamonds in **a** and **c** bracket the timing of the outburst, with filled diamonds representing the last available image before the GLOF and open diamonds the first available image after the GLOF.

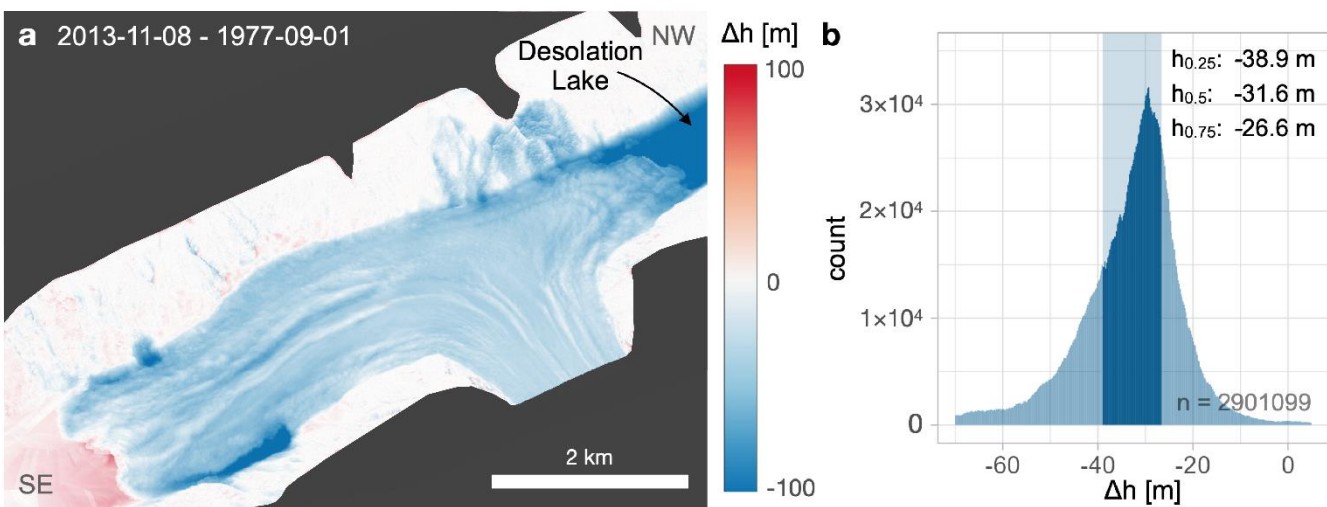

**Fig. 3: Elevation change ($\Delta$h) of Lituya Glacier and surrounding area between 1977-09-01 and 2013-11-08. a**; Oblique 3D view of 2013-11-08 point cloud generated from the ArcticDEM (Porter et al., 2022) coloured by vertical point cloud distance on masked (black) background. Positive and negative values show elevation gains and losses, respectively. Regions with no data are displayed in black. **b**; Histogram of $\Delta$h within the 2013 glacier outline (Fig. 4) with 0.2 m bin width. The dark blue shade marks the interquartile range (IQR; h-

255 values refer to quartiles and median).





**Fig. 4: Elevation time series of Lituya Glacier between 2013 and 2019. a**; Elevation differences of ArcticDEMs (Porter et al., 2022) between 2016-12-10, 2018-09-02, and 2019-04-01, referenced to 2013-11-08. The 2013-11-08 glacier outline for estimating the elevation change (Δh) across the glacier is outlined in black. The high positive values on the proglacial delta in the left panel are an artefact from cloud cover in the 2016 DEM. **b**; Pixel-wise elevation changes within the 2013-11-08 glacier outline using histograms with a bin width of 0.2 m. Highlighted areas in dark blue mark the interquartile range; h-values refer to quartiles and median. **c**; Density plots of the mean annual elevation change rate across Lituya Glacier derived from the difference to the 2013 DEM.

## 4.4 Surface lowering and glacier properties



The surface elevation of Lituya Glacier decreased during the past half-century. Across the glacier dam area, we find

a median surface elevation change of $-31.6_{-38.9}^{-26.6} \pm 4.4$ m (interquartile range in super- and subscript $\pm \sigma_{\Delta h}$) between 1977 and

2013 and a further $-17.5_{-20.7}^{-13.4} \pm 3.1$ m loss between 2013 and 2019, with $-11.7_{-14.6}^{-7.8} \pm 2.8$ m of elevation lost in only three years

(2013-2016) (**Figs. 3, 4**). The average annual rate of surface elevation change tripled from $-0.9_{-1.1}^{-0.8} \pm 0.1$ m yr$^{-1}$ between 1977-

2013 to $-3.2_{-3.8}^{-2.4} \pm 0.6$ m yr$^{-1}$ between 2013 and 2019. Between 2013 and 2019, most parts of the glacier surface dropped 2 to 6

m yr$^{-1}$ (**Fig. 4c**). The elevation loss of the glacier dam corresponds to a mass loss of $-0.8 \pm 0.1$ m w.e. yr$^{-1}$ between 1977-2013

and $-2.9 \pm 0.6$ m w.e. yr$^{-1}$ between 2013-2019. The elevation-change error ($\sigma_{\Delta h}$ calculated over the static surfaces is 4.4 m for

the aligned point clouds (2013-1977) and ranges between 2.8 and 3.8 m for the co-registered ArcticDEMs (2019-2013, 2018-

2013, 2016-2013) (**Fig. S4**). This is only 14-23 % of the median surface elevation change over the respective time period and

thus well below the measured elevation changes across Lituya Glacier dam.

We found heterogeneous surface lowering across the Lituya Glacier dam. Elevation loss was greatest at the lake edge

between 1977 and 2013 (**Fig. S6a**). During this period, local elevation losses within 100 m from the lake edge exceeded 70 m,

which is at least twice the median in this time period. In contrast, between 2013 and 2019, surface lowering of the ice margin

was similar to elsewhere on the dam. Surface lowering was greater near the glacier outlets towards Lituya Bay (**Fig. 1a,S6**).

Between 1977 and 2013, the surface loss was almost twice (>50 m) the median across the whole glacier dam near the eastern

and the western outlets (**Fig. S6a**). Between 2013 and 2019, surface lowering affected mostly the eastern outlet (**Fig. S6b**).

Glacier elevation losses on debris covered areas are similar to those on areas with exposed ice. For example, between 2013-

2019, clean-ice areas lowered by $-3.5_{-0.4}^{+0.4} \pm 0.6$ m yr$^{-1}$ compared to $-3.1_{-0.7}^{+0.8} \pm 0.6$ m yr$^{-1}$ on debris-covered areas.

Surface lowering of Lituya Glacier has been accompanied by several terminal advances and downwastings. Lituya

Bay became landlocked after the glacier terminus advance in 1992. The debris-covered terminus continued to advance and

retreat within a total range of ~100 m on average until 2011. Since 2012, the terminus retreated up to ~400 m, leaving behind

multiple moraines that are <10 m high on the proglacial delta (**Fig. 6b**). The moraines are built from and partially buried by

flood deposits, indicating that the glacier advanced after the delta began to form towards Lituya Bay. The lake terminus of

Lituya Glacier retreated and advanced several times, interrupted by years of stagnation, and the elevation differences mark

advances between 2013 and 2019 (**Fig. 4a**). The net advance of Lituya Glacier towards Desolation Lake was 400 m between

1985 and 2023. Yet, Desolation Lake began to grow southward at its contact with Lituya Glacier in 2003 along the western

valley wall (**Figs. 3, 6a**). Comparing pre- and post-flood images in the following years reveals more evidence for further retreat

of the glacier front at this location.

Our IPR survey in June 2023 revealed striking differences in ice thickness and subglacial topography along Lituya

Glacier (**Fig. 5a**). Ice in the higher northern profile has an estimated maximum thickness of 492 ±20 m (**Fig. 5b**). Along the

northern and southern profiles, the glacier has a mean elevation of 325 m and 280 m, respectively. We infer that more than a

295 third of the cross-sectional area of the glacier is below sea level, with a maximum depth of 158 m below ellipsoid (h.b.e.). The

lower southern profile is characterised by a shallower glacier bed with a maximum depth of 97 m h.b.e. (**Fig. 5c**). Both profiles



are asymmetric, with a more gentle west and a steeper east wall (**Fig. 5b,c**). Thus, the area with the greatest ice thickness and the lowest elevation of the glacier bed is located closer to the eastern valley wall, close to where the glacier enters Desolation Valley and separates into two branches with numerous longitudinal ice ridges covered by debris (**Figs. 5a, 6d**).

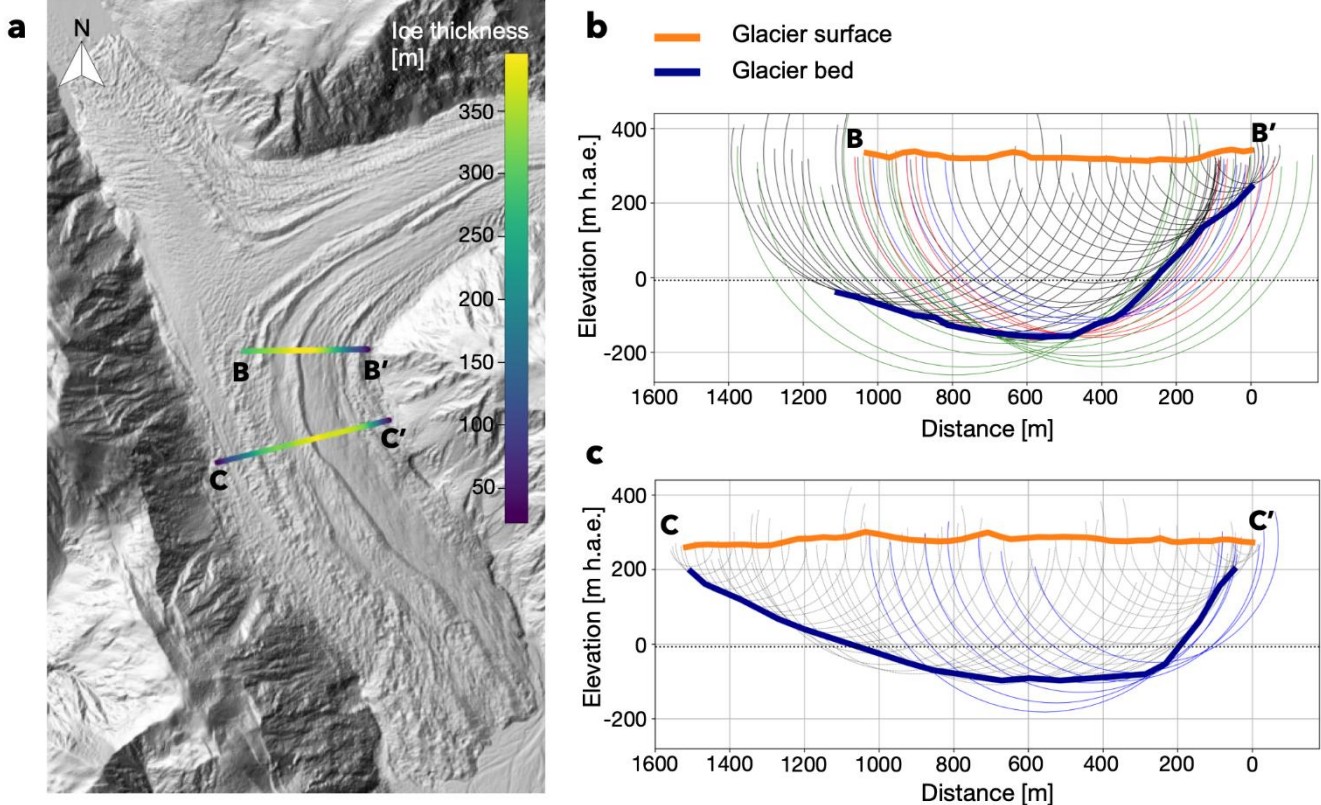

**Fig. 5: Glacier geometry of Lituya Glacier along two profiles derived from an ice-penetrating radar (IPR) survey in June 2023**. a; Location of the IPR profiles plotted on the 2019 ArcticDEM (Porter et al., 2022). The colour along the profiles show the ice thickness. **b,c;** Elevation of the glacier surface (orange) and estimated glacier bed (blue) along the profiles. The dotted line shows mean sea level (EGM96 geoid). The light curves are the return ellipses, black colours indicate first return (**section 3.4**).

**4.4 Drainage characteristics**

While the subglacial drainage network remains unknown, we infer the presence of a channel inlet, through which Desolation Lake drains, from local glacier erosion during the drainages at the western lake edge (**Fig. 6a**). Pulses of sediment flushed into Lituya Bay during the outbursts reveal two different flood outlets since the formation of Desolation Lake, both of them hugging the western and eastern valley walls (**Fig. 1a, 6c**). In the 1970s, only the western outlet was active. However,

between 1981 and 1985, the eastern glacier outlet also flushed sediment into the bay. Yet, the western outlet remained the dominant source of sediment during outbursts until 2007. Between 2007 and 2012, the drainage path shifted. The delta deposits in front of the outlets show a sharp colour contrast (**Fig. 6c**). Since 2012, outwash was delivered primarily to the delta from the eastern outlet, indicating that all floods have been routed through this outlet. The glacier front at the eastern outlet has



locally retreated ~1 km since 1995, decreasing the horizontal distance between the inlet at Desolation Lake and the outlet

towards Lituya Bay to ~4 km in 2023 (**Fig. 3a**). Between 2013 and 2019, outwash and rockfall has accumulated at the base of

the slope where the glacier front retreated with up to 80 m thickness (**Fig. 4a**).

**Fig. 6: Geomorphic evidence of repeated outbursts of Desolation Lake and glacier surface properties of Lituya Glacier. a**; View to
the south of the Lituya Glacier ice dam showing the retreat of the glacier along the west side of the dam. **b**; Field photo of a moraine likely
built from outburst flood deposits in front of the Lituya Glacier terminus. **c**; RapidEye image of the proglacial delta on 29 August 2012, i.e.
10 years prior to the situation in **Fig 1a**, showing the colour contrast between flood deposits. Photo locations of **b** and **d** are marked in **c. d**;
Photo of the glacier surface close to the ice divide of Lituya Glacier.

## 5 Discussion

### 5.1 Surface lowering of Lituya Glacier





Surface lowering of Lituya Glacier since at least 1977 is consistent with the regional trend of glacier-elevation loss in Alaska (Huss, 2013; Trüssel et al., 2013; Larsen et al., 2015). Between 2000 and 2019, the mean regional ice surface lowering was -0.91 m yr$^{-1}$ (Hugonnet et al., 2021). Rates of glacier elevation change differ across the region, depending on factors such as elevation, aspect, hypsometry or terminus type. Lake-terminating glaciers in southern coastal Alaska have the largest elevation change rates in recent decades (Larsen et al., 2015; Yang et al., 2020). When assuming that all surface

lowering is due to the loss of ice, the estimated thinning rate of Lituya Glacier dam between 1977 and 2013 is -0.8 ±0.1 m w.e. yr$^{-1}$, which aligns with the regional median of lake-terminating glaciers in southeastern Alaska near this period (-0.94 m w.e. yr$^{-1}$ from 1994 to 2013; Larsen et al., 2015). However, the available regional rates are hardly comparable with our estimates, considering that they cover different time periods and glacier elevation change, and associated thinning rates have accelerated in past decades (Hugonnet et al., 2021). Consistent with the regional acceleration, the estimated thinning rate of Lituya Glacier

increased to -2.9 ± 0.6 w.e. yr$^{-1}$ over the period 2013-2019. Our estimates are based on only ~11 km$^2$ of the ablation zone and do not estimate the mean ice loss rate of the entire glacier surface. Yet, we consider the elevation changes across the dam region to be most relevant for investigating the influence of local glacier changes on the size and drainages of Desolation Lake.

Considering the subaerial size of the proglacial delta (3 km$^2$) towards Lituya Bay, some surface lowering might be attributed to the removal of sediments at the glacier bed. This effect has been observed for Taku Glacier, a former tidewater

glacier in southeast Alaska; the glacier has entrenched itself into glacimarine sediments by up to 55 m along its centreline within 24 years (1989-2003) (Motyka et al., 2006). At the Lituya Glacier ice divide, our IPR data shows a glacier bed depth of at least ~150 m h.b.e., likely carved by subglacial erosion during previous advances. Sedimentation rates measured in Lituya Bay in the first half of the 20$^{st}$ century suggest that glacial erosion rates could have been as high as 45 mm yr$^{-1}$ for Lituya and Crillon glaciers (Jordan, 1962; Hallet et al., 1996). Subglacial sediment remobilization might further explain the hardly

deviating surface lowering rates across debris-covered regions and areas with exposed ice. In addition, the repeated outburst floods likely mobilized subglacial material along the drainage path that helped form a large proglacial delta. The pronounced lowering of the ice surface near the active outlets (**Fig. S6**) indicate that rates of subglacial sediment removal or the loss of glacier mass might have increased since the lake entered the flood cycle. The floods also likely contribute to the loss of glacier mass due to localized melt from frictional heat produced by the water flow (Liestøl, 1956; Nye, 1976) and evacuation of ice

chunks into the proglacial area. However, even increasing rates of sediment remobilisation and localized erosion of ice and melt along the drainage path cannot explain the acceleration and amount of surface lowering across the entire glacier dam. Thus, large parts of the surface lowering are likely contributed to surface melt, eventually causing the thinning of Lituya Glacier.

## 5.2 Effects of glacier mass loss on Desolation Lake

Glacier thinning in Alaska is well documented (e.g., Motyka et al., 2003; Larsen et al., 2007; Trüssel et al., 2013; Hugonnet et al., 2021), and lake drainage volumes have decreased in this region, locally accompanied by shrinking lake areas (Veh et al., 2023). In contrast, our data show that Desolation Lake has increased both in size and drainage volume between



1985 and 2014. The estimated outburst volumes of Desolation Lake between 2009 and 2014 stand out compared to those of
other ice-dammed lakes in Alaska. The largest flood volumes are more than five times the regional median of $130 \times 10^6$ m$^3$
(Lützow and Veh, 2024) and rank among the largest reported GLOFs from ice-dammed lakes worldwide in the past decade.
Only the outbursts of Lake Tininnilik ($1830 \times 10^6$ m$^3$; Kjeldsen et al., 2017), Cataline Lake ($2500 \times 10^6$ m$^3$; Grinsted et al.,
2017), and Lago Greve ($3700 \times 10^6$ m$^3$; Hata et al., 2022) exceed our largest estimated flood volume of ~725 ±145 $\times 10^6$ m$^3$.

       Our method for estimating lake levels and outburst volumes has residual uncertainties that remain difficult to quantify.
For example, the method only captures the subaerial storage volume of the lake, and thus might miss water in a potential wedge
beneath Lituya and Faiweather glaciers (**see 5.3**). We also saw water-filled crevasses in Fairweather Glacier draining during
the outbursts, suggesting that either the lake extends underneath the glacier or the crevasses are connected with the lake through
subglacial tunnels. We had to exclude water in crevasses because the depth and geometry of the crevasses remain unknown.
However, as the crevasses grew, the drainage volumes were more and more underestimated until Desolation Lake finally
covered the crevassed glacier in 2012. In the years prior (2007-2012), our drainage volume estimates that include the crevassed
region exceed the conservative minima by up to +130%. Therefore, the decreasing drainage volumes between 1985 and 2006
might be largely a consequence of this estimation bias. Furthermore, freshly calved ice may have raised the lake level of
Desolation Lake because water was displaced by icebergs. This process probably effected the lake level the most, albeit by an
unknown quantity, between 2012 and 2014 when parts of Fairweather Glacier broke up into floating icebergs. Landslides from
adjacent slopes also might have changed the lake bathymetry, and thus levels, during our study period. However, the flood
volumes are largely robust against infilling because the lake only drains partially.

       We argue that the topographic and glacier setting explain why Desolation Lake produces larger floods at decreasing
lake levels. For much of our study period, the lake was in contact with two retreating glaciers in a confined valley. Assuming
that the lake bathymetry remains largely unchanged, the lake must expand in size and volume and hence in length or width to
sustain growing water and flood volumes while lake levels gradually drop. Desolation Lake cannot expand much in width due
to the steep adjacent slopes in Desolation Valley. However, the lake was able to grow greatly in length and freed a ~5 km-long
reach of the valley from ice within a few years. Similar observations have been made for Suicide Basin at Mendenhall Glacier,
located ~200 km southeast of Lituya Bay, where the lake volume increased due to progressive deglaciation of the basin
(Kienholz et al., 2020).

How can an ice-dammed lake generate progressively larger floods while its lake level declines at the same time? We
argue that, first, the accommodation space created by the retreat of Fairweather and Desolation glaciers must be larger than
the potential loss of storage volume due to the fall in lake level. Second, the lake needs a growing water source to repeatedly
refill the drained basin. Thus, the increase in both lake size and outburst volumes is likely to halt when retreat of Fairweather
and Desolation glaciers no longer creates sufficient new accommodation space or the water supply per unit accommodation
space decreases. Therefore, lake growth might stall when the lake loses contact with the receding glaciers and the created
accommodation space per unit terminus retreat declines. Accordingly, an increase in drainage volumes over several decades





might be more likely for deep lakes or lakes with a gently sloping lake bed, enabling the lake to stay longer in contact with the retreating glacier.

Water sources that annually refill Desolation Lake after each outburst include glacier melt, snow melt, and rainfall. In our study period, the drainage frequency has changed little over past decades (**Table S2**), while the lake area and outburst volumes grew. Thus, the total water input into Desolation Lake must have increased or the non-catastrophic outflow of the lake, for example through incomplete sealing of the dam, must have declined to enable the lake to refill. Water that remained in the lake during previous outburst floods may also contribute to higher outburst volumes, as Desolation Lake drains to lower levels with time. Average annual precipitation is >5000 mm (Wendler et al., 2017) in the watershed feeding Desolation Lake (~350 km$^2$), and should be sufficient to refill the lake alone (**Fig S7**). However, half of the watershed is covered by Fairweather and Lituya glaciers, both having a second terminus. Thus, some of the rain and glacial meltwater might be unable to reach Desolation Lake. The total precipitation input into the lake is difficult to estimate as some of the precipitation across the Fairweather and Lituya glaciers and surface runoff from the slopes surrounding the glaciated basins may enter the glacial drainage systems, for example through crevasses at the glacier surface, and routed towards other termini. Yet, annual precipitation increased by 8% in southeast Alaska between 1949 and 2016 (Wendler et al., 2017), potentially increasing the overall runoff from both glaciated and unglaciated terrain into the lake.

We believe that the melting of Lituya, Fairweather, and Desolation glaciers is another important source of water to Desolation Lake. While accelerated surface lowering of Lituya Glacier is consistent with regional and global trends (Yang et al., 2020; Hugonnet et al., 2021), the drainage volumes of ice-dammed lakes throughout Alaska have declined (Veh et al., 2023). Yet, the increase in meltwater in response to enhanced glacier surface lowering might have had only minor influence on the growth of the lake. Extrapolating our most recent estimate of surface lowering (-2.9 m ±0.6 w.e. yr$^{-1}$; 2013-2019) over the entire glacier (76.6 km$^2$, RGI 6.0), the annual runoff from surface melt of Lituya Glacier accounts for about one third (~220 x 10$^6$ m$^3$ yr$^{-1}$) of the more recent estimated GLOF volumes. Yet, the real runoff from surface melt into Desolation Lake is likely lower considering that this estimate neglects runoff towards Lituya Bay, overestimation due to extrapolation of surface lowering derived from the ablation zone, and assumes that all surface elevation loss is from glacier thinning which is likely not the case (**see section 5.1**). The estimated thinning rate of Fairweather Glacier (212 km$^2$) is relatively low at -0.4 m w.e. yr$^{-1}$ (2005-2013; Larsen et al., 2015), which is equal to a water volume of ~85 x 10$^6$ m$^3$ yr$^{-1}$. However, the area of Fairweather Glacier that Desolation Lake has claimed is large, being about two times the size of the southern lake before the water bodies merged. Concurrently, the velocity of the arm of Fairweather Glacier flowing into Desolation Valley has increased since 2000 (**Fig. S8**; NASA its-live, https://its-live.jpl.nasa.gov/, last access: 2024-06-01). Hence, much of the water refilling the lake may come from ice loss due to lateral downwasting of Fairweather Glacier and calving at the glacier front rather than surface melt.

While Desolation Lake doubled its area between 2006 and 2014, its growth in length has slowed subsequently. Since 2014, the area of the lake has decreased by 4 km$^2$ while the glacier front of Fairweather Glacier remained largely within a range of ~240 m. Considering the ongoing negative trend in maximum annual lake levels, Desolation Lake may finally be entering a period of declining GLOF volumes. Hence, it may have returned to the regular jökulhlaup cycle (Evans and Clague,





1994). This cycle ends when the ice dam, and hence the lake, disappear and outbursts stop, a fate that has already been reached by other lakes in the region, for example Tulsequah Lake, BC, Canada (Geertsema and Clague, 2005). In the future, Desolation Lake may eventually drain through decaying Fairweather Glacier, considering that the northern part of the lake is wider and possibly deeper. Finally, the lake may disappear entirely when Fairweather or Lituya glaciers shrinks to the point where it can
no longer impound a lake.

## 5.3 Drainage mechanism

Desolation Lake likely drains when the ice dam begins to float on the lake, followed by subglacial tunnel enlargement. Flotation as a flood trigger could explain some of the negative long-term trend in pre-flood lake level tied to a thinning Lituya Glacier (**Fig. 2c**). Ice dams begin to float when the hydrostatic pressure of the water in the lake exceeds the ice overburden
pressure of the glacier (Rabot, 1905; Thorarinsson, 1939; Sturm and Benson, 1985). Therefore, the critical lake-level threshold required to initiate drainage is expected to lower as the ice dam thins (e.g., Marcus, 1960; Sturm and Benson, 1985; Tweed and Russell, 1999). A negative trend in pre-flood lake levels could also be explained by a decrease in ice dam elevation if the lake spills over the glacier at a certain level (e.g., Liestøl, 1956; Dømgaard et al., 2023). During our field visit in May 2023, we found no channels at the glacier margin and a dam height of ~30 m above the lake level (**Fig. S9**), only two weeks before
the lake drained. This observation rules out dam overflow as process limiting lake level or initiating drainage.

The floodwaters likely enter a channelized subglacial drainage system after flood initiation. From the quantity of outwash deposited during floods (**Fig. 6c**), we infer that the drainage channel must follow the glacier bed, so that the floods can entrain sediment. The reasons for the observed shift in drainage paths remains unknown. An explanation might be the geometry of Lituya Glacier, conditioned by the ice divide and frequent glacier front advances as an indicator for phases with
strong ice flow, resulting in the permanent blockage of the western outlet. Dam flotation and subsequent channel enlargement have been inferred for other ice-dammed lakes, including Gornersee, Switzerland (Huss et al., 2007), Hazard Lake, YT, Canada (Clarke, 1982), Hidden Creek Lake, Kennicott Glacier, Alaska (Anderson et al., 2003), and the ice-marginal lake at Russell Glacier, Greenland (Carrivick et al., 2017). Yet, the critical lake level required for dam flotation likely competes with other factors that might explain the annual variability in the lake-level data of Desolation Lake. We observe an intermediate increase
in pre-flood lake levels for three outbursts of Desolation Lake between 2009 and 2014 (**Fig. 2c**). This period coincides with extensive melt of Fairweather Glacier, suggesting that the pre-flood level might have been affected by a period of high meltwater inflow or experienced sudden rise due to water displacement by freshly calved ice. We further observe some drainages at very low pre-flood lake levels, for example in 1999, 2005 and 2006, that deviate from the overall trend between 1985 and 2022, likely initiated by tunnel enlargement alone (**Fig. 2c**). The drainage mechanism of an ice-dammed lake may
change on annual basis as, for example, described for outbursts of Gornersee in Switzerland (Huss et al., 2007). Opening and enlargement of a tunnel may occur before the flotation threshold is reached, for example by melting due to the thermal energy of the lake water and frictional heat of the flow (Liestøl, 1956; Mathews, 1965; Nye, 1976; Clague and O'Connor, 2021). Furthermore, incomplete sealing of the subglacial drainage tunnel due to thermal or mechanical weakening of the ice dam can



lead to earlier tunnel enlargement and hence, drainage of the lake at lower water levels (Dømgaard et al., 2023). Satellite
images show a constant lake level in the five months following the outburst in 2023, indicating temporary incomplete sealing.
Drainage might also be initiated when the water pressure in the en- or subglacial conduit decreases, for example due to a
decline in meltwater production as a result of a drop in air temperatures. A larger pressure gradient between lake and englacial
waters may enable the lake to connect to the drainage system of the glacier (Tweed and Russell, 1999; Russell et al., 2011;
Dømgaard et al., 2023).

Our time series of lake levels shows that the lake partially empties to 85-95% of its initial elevation on average (**Fig.
**7b**). In contrast, other lakes that are hypothesized to drain by ice-dam flotation drain completely, for example Hidden Creek
Lake (Anderson et al., 2003) and Summit Lake, British Columbia (Mathews, 1965). Considering that Desolation Lake always
drained only partially, we suspect that a moraine or bedrock sill is located at a level above the lake bed to prevent the lake
from completely emptying (**Fig. 7c**). Subglacial erosion could have left a bedrock high or, alternatively, provided material for
a terminal moraine that is now buried beneath Lituya Glacier after its advance during the Little Ice Age. Based on these
assumptions, we expect that the lake drains down to the elevation of the sill, or beyond. To allow for the observed negative
trend in the post-flood lake levels, the sill would have to erode over time. Considering that the post-flood lake levels declined
by at least 100 m, remobilisation of unconsolidated material of a purported moraine may be more likely rather than bedrock
erosion.

Where is the seal of the lake located? The simple assumption that the post-flood lake level is equivalent to the
elevation of the glacier bed at the ice dam allows us to explore possible flotation ratios. Following this assumption, the lake
level drop is equivalent to the water depth at which ice dam flotation at the seal is initiated (**Fig. 7c**). We can thus reconstruct
the ice thickness and elevation of the glacier surface at the seal based on different flotation thresholds. We neglect potential
siphoning of water at the seal though a narrow channel that may lead to drainage below the seal elevation and thus would result
in a smaller pre-flood water depth at the seal than derived from the lake level drop during the outburst. As our scenarios are
based on flotation thresholds, a smaller water depth would result in a smaller estimated dam height (i.e. ice thickness) at the
seal. Therefore, the larger the elevation difference between the sill and the post-flood level due to water siphoning, the more
would the estimated location of the seal shift towards the glacier front, characterized by lower glacier surface elevations.

We base our exploration of these ideas on our best data scenario with available lake level and glacier elevation data
at the time of outburst. For the outburst in late October 2013 (**Table S2**, outburst #36), we estimated a lake level drop of 30 m
(i.e. the water depth at the seal) to an elevation of 231 m h.a.e., the elevation of the seal. We explore scenarios in which the
lake level has to reach 60-95% of the dam height to initiate flotation. Assuming thresholds between 80% to 95%, we derive a
glacier elevation between 263 and 269 m h.a.e.. Extracting the respective contour lines from the 2013 DEM, we find that the
seal would have to be located close to the glacier terminus for these scenarios (**Fig. 7a**). Thus, only small parts of the glacier
front would need to float before the outburst began. In contrast, when exploring flotation thresholds between 80% and 95%,
the location of the seal shifts more strongly towards the ice divide, leaving large parts of the dam floating before the threshold
is reached (**Fig. 7a**). We detect ice dam erosion in satellite images showing large-scale detachment of ice along crevasses that



expand parallel to the glacier front (**Fig. S10**). Thus, Desolation Lake may likely extend beneath the ice dam, similar to Hidden
Creek Lake dammed by Kennicott Glacier, Alaska (Walder et al., 2005), Lake Merzbacher, Inylchek Glacier, Kyrgyzstan
(Mayer et al., 2008), and the ice-marginal lake at Kaskawulsh Glacier, Canada (Bigelow et al., 2020). Based on these
observations and the high crevasse frequency towards the lake terminus, we speculate that a flotation threshold of 70-80% is
most likely in case of Desolation Lake despite the partial debris cover. Today, the parallel oriented crevasses extend as far as
the lake eroded into the western margin of the dam (**Fig. S10**). Therefore, the location of seal might have migrated towards the
ice divide of Lituya Glacier in recent years.


**Fig. 7: Exploration of possible flotation ratios at Desolation Lake. a**; Contour lines of Lituya Glacier on hillshade generated from the
2013-08-11 ArcticDEM (Porter et al., 2022). The contours show the reconstructed glacier elevation above the seal for the outburst of
Desolation Lake between October and November 2013 based on different flotation thresholds (60-95%) (section 5.2). **b**; Post- versus pre-
flood levels of Desolation Lake with fitted linear regression (black line) of data colour-coded to timing of the outbursts. The dotted line
shows a 1:1 ratio. **c**; Schematic cross-section of Lituya Glacier along Desolation Valley with simplified subglacial topography below the ice
dam. The marked elevations towards the ice divide are derived from the IPR survey in May 2023. The dotted lines at the ice dam
schematically show the reconstruction of the glacier surface above the seal with different thresholds displayed in **a**.



**Conclusions**

We quantified changes in glacier elevation, lake size, and flood volumes for ice-dammed Desolation Lake, which is growing
in size while releasing GLOFs that partially drain the lake about once per year. We find that the surface elevation of Lituya
Glacier, which dams the lake, declined by ~50 m between 1977 and 2019 with a rate that increased at least three-fold over the
same period. Based on the gradually declining annual lake levels and the amounts of material carried during the floods and
deposited at the glacier terminus, we infer that most outburst floods of Desolation Lake are initiated by flotation of the ice dam
followed by opening of a seal to a channelized path at the bed of the glacier. Based on observations of glacier erosion and
crevasse orientation, we find it plausible that Desolation Lake extends as a water wedge beneath Lituya Glacier and that
flotation happens when the lake level reaches 70-80% of the dam height. Like many other ice-dammed lakes, the levels of
Desolation Lake before the GLOF fell gradually during our study period as a response to ongoing surface lowering of Lituya
Glacier. However, we see an unusual increase in flood volumes over the same period. We argue that the topographic and
glacier setting of the lake, with retreat of the glacier fronts allowing the lake to expand, explains the positive temporal trends
in lake size and outburst volumes with falling lake levels. We argue that ice loss and the increase in accommodation space
following frontal ablation are the key drivers of lake growth, rather than accelerated surface melt. We observed that the
expansion of the lake ceased around 2014, followed by a 4 km² decrease in surface area. In this context, we speculate that
Desolation Lake already has returned to the jökulhlaup cycle with diminishing flood volumes once the increase in lake size
due to lateral expansion no longer outweighs the lowering of the maximum lake level attributable to ice dam thinning. Future
research might focus on verifying the spatial transferability of our observations, for example by investigating ice-dammed
lakes that show similar lateral lake growth in Alaska (**Fig. S1**) and other glaciated mountain regions. Optical satellite images
and elevation models were useful data for reconstructing and quantifying multi-decadal glacier and lake changes. However,
hydrograph data or hourly elevation data acquired form field installations at the ice dam, such as GPS trackers, would
strengthen evidence of drainage initiation and routing. Our insights on the premises of ice-dammed lake growth in a time of
accelerating glacier decay could help identify further sites that might share a similar fate as Desolation Lake. Thus, our study
might pave the way to reveal existing or future lakes that might require careful monitoring, and guide the early implementation
of GLOF risk mitigation measures.

**Code and data availability**

The data and code used to estimate the lake levels and outburst volumes (lake polygons and DEM) as well as the co-registered
DEMs and point clouds, used to analyse glacier elevation changes of Lituya Glacier, are available at
https://doi.org/10.5281/zenodo.13683729.

**Authors contribution**



NL, OK, and GV conceptualized the study. NL led the writing, analysed the data, and compiled the time series of lake and GLOF characteristics. BB and FK contributed to the preparation and processing of the elevation data of Lituya Glacier. MT
collected and post-processed the ice-penetrating radar data of Lituya Glacier. NL, BH, MT, OK, KEH, MG, and GV collected field data at Lituya Bay. All authors contributed to the discussion of the results and the writing of the manuscript.

**Competing interests**

The authors declare that they have no conflict of interest.

**Acknowledgements**

We thank the Glacier Bay National Park for enabling us to collect field data in Lituya Bay and for support during preparation of our field visit. Furthermore, the authors would like to thank Greg Chaney for support during the field stay. We thank Planet Labs Inc. for granting us access to PlanetScope and RapidEye images under the Education and Research programme.

**Financial support**

This research has been supported by the Deutsche Forschungsgemeinschaft (grant no. VE 1363/2-1), the Research Training
Group NatRiskChange (Deutsche Forschungsgemeinschaft, grant no. GRK 2043/2), and KoUP by the University of Potsdam.

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
