# Peer review of "Larger lake outbursts despite glacier thinning at ice-dammed Desolation Lake, Alaska"

_EGUsphere, 2024_

## Author Comment (AC1)

**Referee #1**

We thank the reviewer for their comprehensive and constructive comments on our work. Below, we present their comments in blue font and describe how we plan to address these comments in a revised manuscript in black font. References to specific lines refer to the initial manuscript.

**R1C1**: This is a clearly written paper which explains the somewhat unusual situation of increasing large outburst floods occurring from an ice-dammed lake in Alaska during a period of rapid glacier thinning and retreat. There have been few other studies which have described this unique situation before, and this one was a pleasure to read as it well supported by a variety of comprehensive remote sensing datasets and analyses, and a number of useful figures. I believe that it is worthy of publication, and will add to the growing knowledge of glacier lake outburst floods.

**R1A1**: We thank the reviewer for the positive appraisal of our work.

**R1C2**: My comments are generally very minor, with the exception being the question of whether the lake outburst volume has decreased (perhaps even rapidly?) after 2015. As described in my comments for L425 below, this is a major current limitation with the study as the other datasets for lake area and lake level extend to 2023 (see Fig. 2). If the lake volume estimates could be extended to 2023, and found to be decreasing, then this would change one of the main thrusts of the paper, and perhaps even require the title to be changed from '*larger* lake outbursts'. I encourage the authors to investigate whether other DEM datasets are available that could be used to resolve this question.

**R1A2**: We would like to refer to our detailed reply **R1A15**.

**Individual Comments**

**R1C3**: L23: change to: 'to up to ~700…'

**R1A3**: We will change our wording accordingly.

**R1C4**: L39: I thought that we only had two ice sheets, unless you're calling East and West Antarctica separate ice sheets? Hugonnet et al. (2021) (as well as most other studies) only refer to two ice sheets, the Greenland Ice Sheet and Antarctic Ice Sheet, so I suggest that you specify two here.

**R1A4**: Yes, we were referring to East Antarctica, West Antarctica, and Greenland. However, we agree with the reviewer that Antarctica is more commonly referred to as one ice sheet in the literature. Therefore, we will change the wording as following: (L38-39) "…accounting for a quarter of the total global glacier mass loss outside the *two* ice sheets (Hugonnet et al., 2021)".

**R1C5**: L75: change 'storing' to 'storage'

**R1A5**: We will change our wording accordingly.

**R1C6**: L89: you state that Alaska has 667 glacier lakes, but this is incorrect: as stated in the title of Rick et al. (2022), this total is for Alaska and NW Canada. A pet peeve of mine is that studies such as yours seem to implicitly

assume that Canada is part of Alaska, when clearly this isn't true! Throughout your paper you need to ensure that you properly define which area Alaska refers to, and include Canada when necessary.

**R1A6**: We thank the reviewer for pointing this out. We will now explicitly focus on lakes in Alaska and will change these statistics based on a manually mapped glacier lake inventory by Zhang et al. (2024): (L89) "In 2020, Alaska hosted 1,408 glacier lakes >0.05 km$^2$, 132 of which were ice-dammed lakes (Zhang et al., 2024)."

**R1C7**: L95: figures should be referenced in sequence: you're referencing Fig. 6 here, but haven't even referenced Fig. 1 yet.

**R1A7**: We will adjust the reference sequence in the manuscript accordingly.

**R1C8**: L165: change 'which assume' to 'which are assumed'

**R1A8**: We will change our wording accordingly.

**R1C9**: L176: missing a bracket after $\rho$
**R1C10**: L178: need to enter symbol for area between the brackets
**R1C11**: L180: symbol and bracket missing at start of line

**R1A9-11**: We thank the reviewer for identifying these errors and will correct them accordingly.

**R1C12**: L192: presumably there is also an uncertainty from the change in surface elevation between the acquisition date of the 2019 ArcticDEM and the June 2023 radar survey? Your results suggest that this can amount to several metres per year.

**R1A12**: We thank the reviewer for bringing this to our attention and will add this as following: (L192) "Uncertainties in ice thickness stem from possible deviation from the assumed radar wave speed (±2 m/μs), from the accuracy with which a return can be picked (about 0.1 μs), and from the interpolation from the discrete point measurements. Based on cross-over analysis from similar surveys, we estimate uncertainties to be ±20 m (e.g., Tober et al., 2023). *Uncertainties in bed elevation further stem from the surface elevation extracted from the 2019 DEM. Based on our latest estimate of elevation change (2013-2019), we expect the glacier surface to have further lowered ~13 m on median between 2019 and 2023.*"

**R1C13**: L335: -2.9 requires a unit (presumably m)

**R1C13**: We will add the unit.

**R1C14**: L356 (& L409): some references that are missing here and describe opposite patterns (lake drainage volumes increasing over time) are Kochtitzky et al. (2020) and Painter et al. (2024), who describe how releases from ice-dammed Dań Zhùr (Donjek) Lake, Yukon, have been increasing towards present day, primarily as a result of a larger basin becoming available for the lake to form in as the glacier retreats. These damming events and subsequent releases are primarily controlled by glacier surging, but share some similarities to what is being described in this study and so it seems that they should be mentioned:

Kochtitzky, W., Copland, L., Painter, M. and Dow, C. 2020. Draining and filling of ice dammed lakes at the terminus of surge-type Dań Zhùr (Donjek) Glacier, Yukon, Canada. Canadian Journal of Earth Sciences, 57, 1337-1348

Painter, M., Copland, L., Dow, C., Kochtitzky, W. and Medrzycka, D. 2024. Patterns and mechanisms of repeat drainages of glacier-dammed Dań Zhùr (Donjek) Lake, Yukon. Arctic Science, 10(3), 583-595. https://doi.org/10.1139/as-2023-0001

**R1A14**: We thank the reviewer for bringing these studies to our attention. We will highlight them in our revised discussion: (L380) "However, the lake was able to grow greatly in length and freed a ~5 km-long reach of the valley from ice within a few years. Similar observations have been made for Suicide Basin at Mendenhall Glacier, located ~200 km southeast of Lituya Bay, *and Dań Zhùr Lake at Donjek Glacier, ~300 km northwest of Lituya Bay,* where the lake volume*s* increased due to progressive deglaciation of the basins (Kienholz et al., 2020; *Kochtitzky et al., 2020; Painter et al., 2024*). *However, damming Dań Zhùr Lake followed glacier surges, sealing off the river discharge in the main valley trunk, and thus differs from the case of Suicide Basin and Desolation Lake.*"

**R1C15**: L425: This is a key point that needs to be better investigated. One of the main contentions of this study is that drainage volumes from Desolation Lake have been increasing over time, but no outburst volume estimates are provided after 2015 (Fig. 2b), during a multi-year period when the level of Desolation Lake is rapidly dropping (Fig. 2c). This lack of recent volume estimates seems to be due to the lowest lake level being 197 m in an ArcticDEM from 2020-09-11. My expectation is that the lake volume dropped after 2015, which if quantified would add significantly to the story being presented, and perhaps change some of the final conclusions. The solution to providing outburst volumes after 2015 would be to use a base DEM collected when the lake is at a lower level than 197 m. I'm unsure how much searching the authors have done for datasets beyond the ArcticDEM, but there are several potential sources of DEM information for this period that address this issue:

- The USGS collected LIDAR data over at least part of the lake in summer 2019, which can be downloaded from: https://portal.opentopography.org/usgsDataset?dsid=AK_GlacierBay_B3_2019
- DEMs can be generated for free from stereo ASTER imagery, acquired regularly up to present day: https://lpdaac.usgs.gov/products/ast14demv003/
- Cryosat data has been collected since 2010, and standardized datasets are now available such as CryoTEMPO Inland Water which might provide useful elevation data: http://cryosat.mssl.ucl.ac.uk/tempo/index.html
- IceSAT2 has been operating since 2018, with elevation data from tracks over the lake available to download from locations such as: https://openaltimetry.earthdatacloud.nasa.gov/data/icesat2/

**R1A15**: We agree with the reviewer that extending the time series of drainage volume could strengthen the discussion and conclusions. We also appreciate the new data sources, some of which we were not aware of before. By searching for suitable DEMs, we had compared different products such as the SRTM DEM, the USGS Glacier Bay Lidar product or the ASTER GDEM. We had selected the 2020 strip of the Arctic DEM because it captures the entire lake surface at the lowest observed level. The 2-m resolution of Arctic DEM is a suitable baseline to robustly estimate lake levels along the steep slopes of Desolation Valley.

Unfortunately, we cannot extend the time series of drainage volumes beyond 2015 using the data suggested by the reviewer. In most cases, we found the lake either veiled in clouds, or no DEMs were available after the drainages in the missing years (2016-2023), in which the lake could be captured at a lower water level than 197 m (height above ellipsoid). Due to the progressively lower post-flood lake levels that we show in Fig. S5, we would require an elevation dataset obtained directly after the latest drainage event to extend the time series until the present. We explored all data sources suggested by the reviewer to test if at least some years following 2015 could be assigned a flood volume. The reasons to exclude the individual data sources are as follows:

- **USGS Lidar data:** This dataset does not fully cover the lake and thus does not allow us to calculate further drainage volumes by filling the DEM to estimated lake levels.
- **ASTER imagery:** We are concerned about the suitability of the ASTER-DEMs to estimate the lake levels and flood volumes of Desolation Lake. The quality of ASTER-images and the resulting DEMs is limited due to satellite jitter that might remain even when extensive correction workflows are applied. Due to the ice cover, low-contrast regions are abundant in our study region and these are problematic for jitter correction, as described in detail by Girod et al. (2017). In addition, cloud-free ASTER images of Desolation Lake are sparse and most available data capture the lake above the 197 m lake level of the

Arctic DEM. We found only one image obtained on 2024-05-08 that captures the lake at a level that would allow us to obtain some further estimates between 2016 and 2019. Yet, we remain cautious about the suitability of a DEM created from these data, in addition to our general concerns, as the lake shore was partially covered with snow avalanches or ice at the time of acquisition:

[Figure]

**Aster image of Desolation Lake on 8 May 2024 showing the shore of Desolation Lake partly covered by snow or ice.**

- **Cryosat:** This dataset might be valuable for our future work. However, the water areas in Alaska, except for Mackenzie River, are not part of the Inland Water product; neither in the region "USLakes" nor any of the other available spatial subsets of the data.
- **ICESat-2:** Creating a DEM at a level below 197 m is challenging using the IceSat2 point samples because we need optical satellite images from the same date to first delineate lake contours and then interpolate the Arctic DEM within the lake boundaries. However, in all but one case, the ICESat-2 lake level data were obtained either during a high stand of the lake or during periods when no satellite images were available due to the low solar angle in December. The only suitable pair of ICESat-2 and Planet images from March 2024 shows the lake heavily covered with snow and ice, so mapping the lake boundary was impractical. Therefore, we could not obtain further drainage volumes to quantify the apparent decrease in post-flood lake levels and volumes. Nevertheless, ICESat-2 data provide further information on the lake levels in reference to our mapped outlines. We will include this in the manuscript as follows: (L143) "Between 2016 and 2023, there are no elevation measurements for post-flood levels below the 2020 ArcticDEM lake level, hindering the approximation of outburst volumes. *For this time period, we use ICESat-2 data obtained through the OpenAltimetry Explorer (https://openaltimetry.earthdatacloud.nasa.gov/, last access: 2024-11-11) to track lake level changes of Desolation Lake.*"

In L235, we will add: "The post-flood levels decreased in a similar manner and dropped below the DEM-derived lake level of 197 m h.a.e. in 2016 (**Fig. 2c**), *hindering the approximation of outburst volumes between 2016 and 2023*. Satellite images show that the lake continued to decrease in width in the following years, indicating that the post-flood lake levels have further dropped since *2015* (**Fig. S5**). *In reference to our mapped lake outlines, ICESat-2 data acquired approximately 2 months after a drainage*

*event in summer 2020 show that the post-flood levels dropped below 183 m until then (**Table S3**). Since 2021, post-flood levels have remained below 172 m, accounting for a decrease of at least 25 m within six years. In 2023, the last year of our record, the pre-flood lake level remained below the DEM lake level of 197 m h.a.e..*"

Furthermore, we will add to our conclusions: (L521) "*A lack of suitable data prevent us from making volume estimates after 2016, leaving open the possibility that lake volumes have recently decreased. Yet,* we observed that the expansion of the lake ceased around 2014, followed by a 4 km² decrease in surface area. In this context, we speculate that Desolation Lake already has returned to the jökulhlaup cycle with diminishing flood volumes (…)."

**Table S3: Lake levels of Desolation Lake acquired by ICESat-2.**

| Date | Lake level |
| --- | --- |
| 2019-03-31 | 195 m |
| 2019-09-29 | 188 m |
| 2020-03-28 | 209 m |
| 2020-03-29 | 209 m |
| 2020-06-27 | 218 m |
| 2020-09-26 | 183 m |
| 2021-12-24 | 172 m |
| 2022-12-22 | 172 m |
| 2022-12-23 | 172 m |
| 2024-03-21 | 176 m |

**R1C16**: L446: dam flotation and subsequent channel enlargement has also been invoked as the casual mechanism for floods from Donjek Glacier by Painter et al. (2024)

**R1A16**: We will include this case in L446: "Dam flotation and subsequent channel enlargement have been inferred for other ice-dammed lakes, including Gornersee, Switzerland (Huss et al., 2007), Hazard Lake, YT, Canada (Clarke, 1982), Hidden Creek Lake, Kennicott Glacier, Alaska (Anderson et al., 2003), and the ice-marginal lakes at Russell Glacier, Greenland (Carrivick et al., 2017) *and Donjek Glacier, YT, Canada (Painter et al., 2024)*."

**R1C17**: Fig. S8: this graph is noisy, presumably because you included all available image-pair velocities no matter their separation time? You should be able to reduce this noise by removing image pairs with short and long similar separation times. For example, I used the ITS_LIVE widget tool (https://itslive-dashboard.labs.nsidc.org/) to only plot data with separation intervals of 30-300 days for your location, which produced a cleaner signal than yours.

**R1A18**: We thank the reviewer for this suggestion. We will replace the figure and now only include data with separation intervals of 100 and 200 days, for which we find a significant reduction of noise compared to the previous graph including all image pairs:

[Figure]

**Fig. S8: Ice velocity of Fairweather Glacier at 58.8078°, -137.6848° between 1984 and 2022 extracted from the ITS_LIVE dataset ([https://its-live.jpl.nasa.gov/](https://its-live.jpl.nasa.gov/), last access: 2024-11-18)**. To reduce noise, we only use data obtained in separation intervals of 100 to 200 days.

**References**

Girod, L., Nuth, C., Kääb, A., McNabb, R., and Galland, O.: MMASTER: Improved ASTER DEMs for elevation change monitoring, Remote Sensing, 9, https://doi.org/10.3390/rs9070704, 2017.

Zhang, T., Wang, W., and An, B.: Heterogeneous changes in global glacial lakes under coupled climate warming and glacier thinning, Communications Earth & Environment, 5, 374, https://doi.org/10.1038/s43247-024-01544-y, 2024.

---

## Author Comment (AC2)

**Referee #2**

We thank the reviewer for their comprehensive and constructive comments on our work. Below, we present their comments in blue font and describe how we plan to address these comments in a revised manuscript in black font. References to specific lines refer to the initial manuscript.

**R2C1**: The authors present a detailed overview of glacial lake outbursts from Desolation Lake (Alaska) using an interesting mix of methods and data. I have a few suggestions for minor adjustments but overall this reads very well, the contents are compelling and I look forward to seeing this published in TC.

**R2A1**: We thank the reviewer for the support for publication of our work.

**R2C2**: I share RC1's view that it would be beneficial to extend the time series of outburst volume beyond 2016 if possible. If the data situation does not allow for this it would be informative to state that somewhere in the manuscript and note why DEM data sources other than the Arctic DEMs were not used. In this case the discussion should be adjusted to more clearly indicate the limitations of the data set (and its interpretation) for recent years.

**R2A2**: We refer the reviewer to our reply **R1A15** that we copy for reference below. In addition, we will emphasize the suitability of the ArcticDEM in the manuscript as following: (L127) "We estimated outburst volumes (…) using the mapped lake area outlines and the 2-m resolution ArcticDEM from 2020-09-11. This DEM (digital elevation model) shows the lake at the *lowest observed* water level of 197 m h.a.e. (height above WGS84 ellipsoid) *among all available DEM products in our study region*. It further has the smallest glacier extent within the ArcticDEM time series (**Table S1**)."

R1A15: We agree with the reviewer that extending the time series of drainage volume could strengthen the discussion and conclusions. We also appreciate the new data sources, some of which we were not aware of before. By searching for suitable DEMs, we had compared different products such as the SRTM DEM, the USGS Glacier Bay Lidar product or the ASTER GDEM. We had selected the 2020 strip of the Arctic DEM because it captures the entire lake surface at the lowest observed level. The 2-m resolution of Arctic DEM is a suitable baseline to robustly estimate lake levels along the steep slopes of Desolation Valley.

Unfortunately, we cannot extend the time series of drainage volumes beyond 2015 using the data suggested by the reviewer. In most cases, we found the lake either veiled in clouds, or no DEMs were available after the drainages in the missing years (2016-2023), in which the lake could be captured at a lower water level than 197 m (height above ellipsoid). Due to the progressively lower post-flood lake levels that we show in Fig. S5, we would require an elevation dataset obtained directly after the latest drainage event to extend the time series until the present. We explored all data sources suggested by the reviewer to test if at least some years following 2015 could be assigned a flood volume. The reasons to exclude the individual data sources are as follows:

- **USGS Lidar data:** This dataset does not fully cover the lake and thus does not allow us to calculate further drainage volumes by filling the DEM to estimated lake levels.
- **ASTER imagery:** We are concerned about the suitability of the ASTER-DEMs to estimate the lake levels and flood volumes of Desolation Lake. The quality of ASTER-images and the resulting DEMs is limited due to satellite jitter that might remain even when extensive correction workflows are applied. Due to the ice cover, low-contrast regions are abundant in our study region and these are problematic for jitter correction, as described in detail by Girod et al. (2017). In addition, cloud-free ASTER images of Desolation Lake are sparse and most available data capture the lake above the 197 m lake level of the Arctic DEM. We found only one image obtained on 2024-05-08 that captures the lake at a level that would allow us to obtain some further estimates between 2016 and 2019. Yet, we remain cautious about the suitability of a DEM created from these data, in addition to our general concerns, as the lake shore was partially covered with snow avalanches or ice at the time of acquisition:

[Figure]

**Aster image of Desolation Lake on 8 May 2024 showing the shore of Desolation Lake partly covered by snow or ice.**

- **Cryosat:** This dataset might be valuable for our future work. However, the water areas in Alaska, except for Mackenzie River, are not part of the Inland Water product; neither in the region "USLakes" nor any of the other available spatial subsets of the data.
- **ICESat-2:** Creating a DEM at a level below 197 m is challenging using the IceSat2 point samples because we need optical satellite images from the same date to first delineate lake contours and then interpolate the Arctic DEM within the lake boundaries. However, in all but one case, the ICESat-2 lake level data were obtained either during a high stand of the lake or during periods when no satellite images were available due to the low solar angle in December. The only suitable pair of ICESat-2 and Planet images from March 2024 shows the lake heavily covered with snow and ice, so mapping the lake boundary was impractical. Therefore, we could not obtain further drainage volumes to quantify the apparent decrease in post-flood lake levels and volumes. Nevertheless, ICESat-2 data provide further information on the lake levels in reference to our mapped outlines. We will include this in the manuscript as follows: (L143) "Between 2016 and 2023, there are no elevation measurements for post-flood levels below the 2020 ArcticDEM lake level, hindering the approximation of outburst volumes. *For this time period, we use ICESat-2 data obtained through the OpenAltimetry Explorer (https://openaltimetry.earthdatacloud.nasa.gov/, last access: 2024-11-11) to track lake level changes of Desolation Lake."*

In L235, we will add: "The post-flood levels decreased in a similar manner and dropped below the DEM-derived lake level of 197 m h.a.e. in 2016 (**Fig. 2c**), *hindering the approximation of outburst volumes between 2016 and 2023*. Satellite images show that the lake continued to decrease in width in the following years, indicating that the post-flood lake levels have further dropped since *2015* (**Fig. S5**). *In reference to our mapped lake outlines, ICESat-2 data acquired approximately 2 months after a drainage event in summer 2020 show that the post-flood levels dropped below 183 m until then (**Table S3**). Since 2021, post-flood levels have remained below 172 m, accounting for a decrease of at least 25 m within six years. In 2023, the last year of our record, the pre-flood lake level remained* below the DEM lake level of 197 m h.a.e.."

Furthermore, we will add to our conclusions: (L521) "*A lack of suitable data prevent us from making volume estimates after 2016, leaving open the possibility that lake volumes have recently decreased. Yet,* we observed that the expansion of the lake ceased around 2014, followed by a 4 km² decrease in surface area. In this context, we speculate that Desolation Lake already has returned to the jökulhlaup cycle with diminishing flood volumes (…)."

**Table S3: Lake levels of Desolation Lake acquired by ICESat-2.**

| Date | Lake level |
| --- | --- |
| 2019-03-31 | 195 m |
| 2019-09-29 | 188 m |
| 2020-03-28 | 209 m |
| 2020-03-29 | 209 m |
| 2020-06-27 | 218 m |
| 2020-09-26 | 183 m |
| 2021-12-24 | 172 m |
| 2022-12-22 | 172 m |
| 2022-12-23 | 172 m |
| 2024-03-21 | 176 m |

Line by line comments:

**R2C3**: L36-39: I would suggest "glacier mass loss" instead of "glacier thinning" as a more general term but this is just semantics.

**R2A3**: We will adjust the wording accordingly.

**R2C4**: I stumbled over "three ice sheets" - don't we usually count two?

**R2A4**: We will adjust the wording as follow: (L39) "…outside the *two* ice sheets", in line with a similar comment raised by reviewer 1 (**R1A4**).

**R2C5**: L112 *We reconstructed the outburst chronology of Desolation Lake and Lituya Glacier between 1882 and 1969 from historic air photos taken by Austin Post and from historic maps (see Supplementary Table S1). From satellite images acquired between 1972 and 2023….*

Can you comment on whether you think you have missed outbursts? How comprehensive is the time series? I understand you can't know if the data isn't there but consider adding a short note on this, mentioning, e.g., changes in temporal coverage with new satellite missions, cloud cover issues (I assume clouds are a major limiting factor?), etc.

**R2A5***:* Indeed, an unknown number of drainages might have gone unnoticed in our analysis. Therefore, we state in the manuscript (L214) "At least 48 outbursts interrupted the growth of Desolation Lake…". The reviewer is right that cloud cover is a limiting factor, in addition to a limited number of satellite images particularly early in our time series when Landsat was the only continuously operating optical satellite mission covering our study region. However, we are confident that we captured most lake outbursts. We visually tracked the lake levels following each GLOF and found that usually more than half a year passes until Desolation Lake rises to a level close to the last pre-GLOF level. The highest increases in lake level occur during the melt season in early summer (May, June, July) when also satellite image availability increases, following limited image availability in winter. To this end, we state in the manuscript: (L216) "Estimating the timing of GLOFs in winter is challenging because clouds and ice commonly cover Desolation Lake for several months, while the solar illumination, and thus

contrast in shadowed regions, decreases". However, we agree that these issues could be emphasized more. We will add a supplementary figure on satellite image abundance in our study region. We will also extend the Methods on the outburst chronology (L122): "To quantify lake-area changes, we manually mapped the lake outlines using the last available cloud-free image before and the first image after each outburst identified from the satellite images with QGIS (…). *While operation periods of optical satellite missions overlap, frequent cloud cover restricted the number of suitable images in the time series. In our study region, only 20-40% of all available images of each Landsat product series show less than 30% cloud cover. Yet, we were able to obtain multiple images in each year except 1983 and 1991 (Fig. S2)."*

[Figure]

**Fig. S2: Number of optical satellite images with less than 30% cloud cover covering our study region between 1972 and 2023.** Colours distinguish optical satellite missions.

**R2C6**: *L173 Lituya Glacier elevation changes were calculated within the reference glacier outline on 2013-11-08 with an area of 10.6 km2 , covering the part of the ablation zone of the glacier that dams Desolation Lake, in the following referred to as Lituya Glacier dam. This outline was cropped above the ice divide to ensure equal coverage of the elevation data (Fig 4)*

I suggest moving this sentence to the beginning of the section to clarify that you computed elevation change only for the lower part (Lituya Glacier dam) rather than the entire glacier. This is important to know for the error estimation and assumptions re. density conversion and it was initially not clear to me. Also: What do you mean by "reference glacier outline"? Did I miss this? Why is that particular outline considered the reference?

**R2A6**: We will move the paragraph to the beginning of the section (L146), so the reference to the glacier outline will be clearer. The revised paragraph will read as: "We calculated glacier *elevation changes only for the dam area (10.6 km$^2$) located in Desolation Valley that we mapped from the 2013-11-08 DEM. We cropped the glacier area north of the ice divide to ensure equal coverage of the elevation data and therefore only refer to the effective dam of Desolation Lake, in the following referred to as Lituya Glacier dam."*

We used the 2013 glacier outline as a "reference" outline for our zonal statistics because we calculated the elevation changes between each DEM and the 2013 DEM that we further used as base for the co-registration.

**R2C7**: Figure 1: The authors have done a good job of presenting their wealth of data in a concise way so I think there is room to consider adding another figure. Splitting Fig. 1 into two figures would allow readers unfamiliar with the site (like myself) to have a very simple overview map at the start of the study site description. Fig. 1 currently is quite crowded and contains a lot of information. I found myself going back to Fig. 1 multiple times while reading looking for features like the ice divide mentioned later on. These things eventually become clear

but I feel that the current panel a in Fig 1 (oblique photo) next to a simple, north oriented, birds-eye map and the inset showing the location in Alaska would be helpful. The other panels (historical maps and satellite imagery) are also valuable but in my opinion could be in a separate figure, which could be introduced a bit later.

**R2A7**: We thank the reviewer for their suggestion and will split this figure accordingly:

[Figure]

**Fig 1.: Location of Desolation Lake. a**; Location map and glacial setting of Desolation Lake. **b**; Oblique aerial view into Desolation Valley from the head of Lituya Bay. Desolation Lake is only partially filled. White arrows indicate the location of the two flood outlets.

[Figure]

**Fig.2: Formation of Desolation Lake.** Time series of georeferenced historic maps (**a-d**; U.S. Coast Survey, 1882; Bien, 1903; U.S. Geological Survey, 1951, 1961) and Landsat images (**e-j**; derived from the USGS Earth Explorer and Google Earth Engine Data Catalog) showing the formation and growth of Desolation Lake between 1882 and 2020. The yellow outlines in **g-j** show the mapped lake areas ($a_L$). The photograph by Austin Post inset into **d** is a view into Desolation Valley and Lituya Glacier from the head of Lituya Bay in 1969, shortly before modern Desolation Lake formed.

**R2C10**: General comment: Figures are not always introduced in the order of their numbering. Adjusting this might make things easier to follow.

**R2A8**: We will rearrange the order of figures in the manuscript accordingly.

**R2C9**: L177  *($\rho$ of 900 kg m-3 (Huss, 2013). We estimated the loss of ice volume within the 2013 reference polygon by multiplying the polygon area () with the median elevation change ($\Delta h$0.5 ). We estimated the total mass loss error ($\sigma\Delta M$) from two normalized error components; the total elevation change error ($\sigma\Delta h$) derived from equation (2), and the mapping error of the 2013 glacier outline ), (adjusted from Shean et al., 2020):*

Some issues here with the parentheses - some are not closed, "area ()" is empty.

**R2A9**: We will correct these errors, thank you.

**R2C10**: L195 does the stated uncertainty include the time difference between the acquisition of the radar measurement and the DEM?

**R2A10**: We would like to refer the reviewer to a similar comment made by reviewer 1 (**R1A12**). We copy our response for reference below:

R1A12: We thank the reviewer for bringing this to our attention and will add this as following: (L192) "Uncertainties in ice thickness stem from possible deviation from the assumed radar wave speed (±2 m/µs), from the accuracy with which a return can be picked (about 0.1 µs), and from the interpolation from the discrete point measurements. Based on cross-over analysis from similar surveys, we estimate uncertainties to be ±20 m (e.g., Tober et al., 2023). *Uncertainties in bed elevation further stem from the surface elevation extracted from the 2019 DEM. Based on our latest estimate of elevation change (2013-2019), we expect the glacier surface to have further lowered ~13 m on median between 2019 and 2023.*"

**R2C11**: L214: Table S2: as mentioned above, can you comment on whether the relatively low number of detected outbursts prior to 1985 is related to a lack of cloud free satellite imagery? Do you have enough images from those years to determine that there were no outbursts?

**R2A11**: The lower number of drainages observed earlier in our record may partly reflect the limited availability of cloud-free images from the Landsat 1-3 missions. However, the annual counts of available images are comparable to the period between 1985 and 2010 as we will show in a new supplementary figure (see our reply **R2A5**). Leaving data availability aside, a further explanation for the lower drainage frequency prior to 1985 might be that Desolation Lake was still in process of formation and only existed as water stored in enlarging crevasses at this time. Therefore, we expect the drainage mechanism to be different from the later drainages when the lake formed a coherent water body around 1985. We will add this to the manuscript as follows: (L454) "The drainage mechanism of an ice-dammed lake may change on an annual basis as, for example, described for outbursts of Gornersee in Switzerland (Huss et al., 2007). *We expect the drainage mechanism to be different from the present lake when Desolation Lake only existed as water stored in enlarging crevasses. The four detected drainages prior to 1985 were likely initiated by subglacial tunnel enlargement. After the lake formed a coherent water body*, opening and enlargement of a tunnel may occur before the flotation threshold is reached, for example by melting due to the thermal energy of the lake water and frictional heat of the flow (…)".

**R2C12**: L235 *The post-flood levels decreased in a similar manner and dropped below the DEM-derived lake level of 197 m h.a.e. in 2016 (Fig. 2c). Satellite images show that the lake continued to decrease in width in the following years, indicating that the postflood lake levels have further dropped since then (Fig. S5)*

Missing word? "..show that the lake level continued to decrease"?

**R2A12**:  Here, we are referring to the post-flood lake width instead of the lake level. We conclude from this observation that the post-flood levels have declined in the following years. We see that the phrasing might have been misleading. We will therefore adjust the manuscript as follows: (L235) "The post-flood levels decreased in a similar manner and dropped below the DEM-derived lake level of 197 m h.a.e. in 2016 (Fig. 2c). Satellite images show that the  width *of the remaining lake after each drainage* continued to decrease  in the following years, indicating that the post-flood lake levels have further dropped since then (Fig. S5)."

**R2C13**: It would be helpful to add a sentence or two here that clearly explains why the time series in Fig 2 b and c stop in 2016 (no DEM showing lake levels below 197m?)

In the methods you state: "*We estimated outburst volumes...using the mapped lake area outlines and the 2-m resolution ArcticDEM from 2020-09-11. This DEM (digital elevation model) shows the lake at the minimum water level at 197 m h.a.e. (height above WGS84 ellipsoid) and has the smallest glacier extent within the available ArcticDEM time series (Table S1). ….Between 2016 and 2023, there are no elevation measurements for post-flood levels below the 2020 ArcticDEM lake level, hindering the approximation of outburst volumes*"

I suggest mentioning this again in the section discussing the results in Fig 2 to make it easier for readers to follow.

**R2A13**: We thank the reviewer for this suggestion and will add this information to the result section as follows: (L235) "The post-flood levels decreased in a similar manner and dropped below the DEM-derived lake level of 197 m h.a.e. in 2016 (**Fig. 2c**), *hindering the approximation of outburst volumes between 2016 and 2023*."

**R2C14**: Fig 4, caption: The high positive values on the proglacial delta in the left panel are an artefact from cloud cover in the 2016 DEM

I suggest marking this in the Fig. like you did for the other features

Panel a, right panel: is the error for "sediment deposition" that points south correct? The deposition is not very apparent. (I think this is explained later in Sec 4.4. but I find it hard to see in the figure)

**R2A14**: We will mark the artefact in the figure accordingly:

[Figure]

**Fig. 4: Elevation time series of Lituya Glacier between 2013 and 2019. a**; Elevation differences of ArcticDEMs (Porter et al., 2022) between 2016-12-10, 2018-09-02, and 2019-04-01, referenced to 2013-11-08. The 2013-11-08 glacier outline for estimating the elevation change (Δh) across the glacier is outlined in black. The high positive values on the proglacial delta in the left panel are an artefact from cloud cover in the 2016 DEM. **b**; Pixel-wise elevation changes within the 2013-11-08 glacier outline using histograms with a bin width of 0.2 m. Highlighted areas in dark blue mark the interquartile range; h-values refer to quartiles and median. **c**; Density plots of the mean annual elevation change rate across Lituya Glacier derived from the difference to the 2013 DEM.

Yes, the arrow points correctly. We agree that the deposition is not very apparent compared to the deposition in front of the outlet. Nevertheless, we find this deposition noteworthy because it is in the order of meters during our study period and shows the built up of the proglacial delta.

**R2C14**: L270 ($\sigma\Delta h$ Close the parenthesis

**R2A15**: We will add the parenthesis accordingly.

**R2C16**: L292 define IPR

**R2A16**: We will add (L292): "Our *ice-penetrating radar (*IPR*)* survey in June 2023…"

**R2C17**: *L335 Our estimates are based on only ~11 km² of the ablation zone and do not estimate the mean ice loss rate of the entire glacier surface. Yet, we consider the elevation changes across the dam region to be most relevant for investigating the influence of local glacier changes on the size and drainages of Desolation Lake.*

I think this warrants a bit more explanation. Why do you consider the elevation change in your investigated section the most relevant? Is there a reason other than it being closest to the lake?

**R2A17**: According to previous studies, the dam area is key to understanding if, and how much, the storage capacity of Desolation Lake changes due to glacier surface lowering, which we had originally included in the introduction: (L73-76) "Despite the global increase in glacier lake volume, this trend towards smaller lakes and floods is also observed for most single ice-dammed lakes with recurring outburst floods and has been attributed to thinning of the local ice dam, limiting the storing capacity of the lake (Evans and Clague, 1994; Tweed and Russell, 1999; Geertsema and Clague, 2005; Shugar et al., 2020; Zhang et al., 2024)."

Nevertheless, the mass balance of the entire glacier is relevant for the water balance of the lake. We had originally discussed this issue in L411-417 and will rephrase the according paragraph to emphasize this more: "Extrapolating our most recent estimate of surface lowering (-2.9 ±0.6 m w.e. yr$^{-1}$; 2013-2019) over the entire glacier (76.6 km², RGI 6.0), *the annual water volume produced by the additional surface melt due to volume loss of Lituya Glacier* accounts for about one third (~220 x 106 m³ yr$^{-1}$) of the more recent estimated GLOF volumes. However, *not all of that water enters Desolation Lake*, considering that this estimate neglects runoff towards Lituya Bay, overestimates the surface lowering due to extrapolating estimates derived from the ablation zone, and assumes that all surface elevation loss is from glacier thinning which is likely not the *case due to significant bed erosion* (see section 5.1)."

**R2C18**: *L352 Thus, large parts of the surface lowering are likely contributed to surface melt, eventually causing the thinning of Lituya Glacier*

This reads a bit odd, maybe change "contributed" to "attributed" or rephrase

**R2A18**: We will adjust the wording accordingly.

**R2C19**: L362 I suggest stating where these lakes are (countries)

**R2A19**: We will adjust the wording as following: (L362) "Only the outbursts of Lake Tininnilik (1830 × 10⁶ m³; Kjeldsen et al., 2017) *and* Cataline Lake (2500 × 10⁶ m³; Grinsted et al., 2017) *in Greenland*, and Lago Greve (3700 × 10⁶ m³; Hata et al., 2022) *in Chile* exceed our largest estimated flood volume…"

**R2C20**: *L423 while the glacier front of Fairweather Glacier remained largely within a range of ~240 m.* Range of what? Is this in relation to some reference point?

**R2A20**: Here we refer to the maximum distance between all terminus positions between 2014 and today. We will adjust the wording as following: (L422) "Since 2014, the area of the lake has decreased by 4 km² while the *terminus* of Fairweather Glacier *advanced and retreated within a small range of ~240m*."

**R2C21**: *L521 We argue that ice loss and the increase in accommodation space following frontal ablation are the key drivers of lake growth, rather than accelerated surface melt.*

This sounds like the two factors are mutually exclusive. Are they? I find it a bit confusing to use the phrases "ice loss" and "accelerated surface melt" as two seemingly contrasting processes. Perhaps add a sentence or two to qualify, or consider rephrasing. "Glacier retreat" would indicate both ice loss and the increase in accommodation

space for the lake (=space where the glacier no longer is). Alternatively, you might focus the explanation more on the difference between frontal ablation and surface ablation.

**R2A21**: Here we were referring to the ice loss from both types of ablation. Yet, we agree that our wording might have been misleading and will rephrase the paragraph as following: (L521) "We argue that frontal ablation is the key driver of lake growth, rather than accelerated surface ablation."